# Provision of Public Dental Services During the COVID-19 Pandemic: Experiences of Dental Staff in Greater Western Sydney, Australia

**DOI:** 10.3390/ijerph21111451

**Published:** 2024-10-31

**Authors:** Tiffany Patterson-Norrie, Ariana Kong, Albert Yaacoub, Ravi Srinivas, Shwetha G. Kezhekkekara, Ajesh George

**Affiliations:** 1Australian Centre for Integration of Oral Health (ACIOH), School of Nursing & Midwifery, Western Sydney University, Liverpool, NSW 1871, Australia; tiffany.patterson@westernsydney.edu.au (T.P.-N.); ariana.kong@westernsydney.edu.au (A.K.); albert.yaacoub@health.nsw.gov.au (A.Y.); ravi.srinivas@health.nsw.gov.au (R.S.); s.kezhekkekara@westernsydney.edu.au (S.G.K.); 2Ingham Institute for Applied Medical Research, Liverpool, NSW 1871, Australia; 3Oral Health Services, Nepean Blue Mountains Local Health District, Penrith, NSW 2750, Australia; 4School of Dentistry, Faculty of Medicine and Health, The University of Sydney, Sydney, NSW 2010, Australia; 5Oral Health Services, South Western Sydney Local Health District, Liverpool, NSW 2170, Australia; 6Translational Health Research Institute, Western Sydney University, Campbelltown, NSW 2560, Australia; 7Faculty of Science, Medicine and Health, University of Wollongong, Wollongong, NSW 2522, Australia

**Keywords:** COVID-19, dentistry, dental officers, oral health, qualitative, teledentistry

## Abstract

Background and aim: The coronavirus (COVID-19) pandemic led to significant changes in health service delivery. Despite the risk in high-exposure environments, frontline workers such as dental staff were expected to continue delivering essential services. This study specifically sought to explore the experiences of dental staff in New South Wales and determine their perceptions of dental care delivery during a pandemic. Methods: Purposive sampling was used to recruit twenty-four dental staff from two local health districts. A deductive framework, as described by Braun and Clark, was used to analyse the transcripts. Four major focus areas were identified: responding to protocol changes, adapting to the impact of changes in policy and protocol, modifying dental treatment planning and recommendations for training and implementation of policies. Results: Dental staff reported that management staff were doing all they could and were most supported working in a team. Reduced contact with patients and personal protective equipment helped staff feel safe when seeing patients. Mental health and remote dentistry services could be more supported. Conclusions: Unique challenges were experienced by dental staff and their management during the global COVID-19 pandemic. Future considerations include improved support for staff and further investigation into the effectiveness of options such as teledentistry.

## 1. Introduction

The novel coronavirus pandemic, also known as COVID-19, is a highly infectious disease that was first recognised in December 2019 [1]. Having a rapid transmission rate, COVID-19 spread globally, affecting more than 156 countries and leading to excessive morbidity and mortality [2]. Common symptoms associated with COVID-19 include gastrointestinal symptoms, fever, weakness, taste and smell disorders, and respiratory symptoms such as sneezing and coughing [2,3]. As knowledge of COVID-19 improved, it became known that the disease is easily transmitted through human-to-human contact, where droplets from an infected person are spread to others by sneezing or coughing or by touching objects that have these droplets [2,4]. To mitigate the rapid transmission of COVID-19, many countries implemented lockdown restrictions to control the spread through means of physical distancing [5].

The severity of illness from COVID-19 is reported to be worse among older people and people with underlying medical conditions [6,7]. Many individuals who are vulnerable to COVID-19, such as the elderly, pregnant women, and children, are also population groups that are noted to be at higher risk of dental problems that may require treatment [8,9,10,11,12]. The acute illness caused by COVID-19, along with the treatments used to combat it, could also lead to a negative impact on oral health [13]. Impaired taste buds, unspecific ulcers, and desquamative gingivitis are some oral signs and symptoms [13]. However, it remains unclear if these symptoms are a common clinical pattern due to the virus itself or a systematic repercussion, considering the weak immune response and negative effects of the treatment [13].

Due to the spread and risk of COVID-19 to both patients and dental staff, procedures or equipment that aerosolised saliva and blood, which have the potential to transmit infected droplets, were generally restricted during the pandemic [14]. Once it was established that the disease could be transmitted via inhalation, ensuring respiratory protection became critically important for the safety of both patients and dental practitioners [15]. Personal protective equipment (PPE) was recommended to all healthcare providers, including dentists, while practicing urgent cases. Research has shown that filtering facepiece respirators (FFRs), including N95, are more effective than surgical masks. However, reusable elastomeric respirators (RERs) were highly recommended over disposables based on their protection factor [15].

Despite dental practice restrictions being adopted by many countries during the pandemic [16], many dental practitioners remained anxious about the potential risk of infection. A recent global study found that 78% of dentists were anxious about the impact of COVID-19 despite having high knowledge levels about recommended practices and changes in protocol to improve infection control [17]. Further, the fear of contracting COVID-19 appears to be higher among dentists who have lower self-efficacy or have an underlying medical condition [18].

Another study in Jordan found that while dentists were aware of the guidelines and mode of transmission of COVID-19, dentists had limited understanding of the additional measures taken to further protect dentists and their patients [19]. Additionally, another problem facing dental and medical practitioners in Australia was the initial worldwide shortage of appropriate PPE designed to reduce the risk of transmission of infectious diseases [20]. With this shortage, some dental staff may experience significant psychological distress about treating patients. A study revealed that Polish dentists experienced a higher rate of coronavirus infection and suffered additional stress due to prolonged exposure to difficult working circumstances and no governmental lockdown of dental offices [21]. Moreover, they also faced shutting down their practice temporarily due to limited protective supplies [21]. A study was carried out to assess the working environment of dental professionals in Poland and Turkey during COVID-19, where it was revealed that the treatment time had increased with each patient with higher usage of protective equipment [22]. In Turkey, usage of reusable respirators was comparatively high, whereas in Poland, a 25% drop in dentists’ revenue was recorded [22]. Several elements were identified in Brazil that need to be considered to prevent burnout among dentists, which include quality of life, length of time in the field, and professional support [23].

Patients were also impacted because of these restrictions, which caused numerous dental treatments to be delayed. When the service was restored, those who needed early treatment had to undergo complex procedures [24,25]. To overcome these challenges, dental practitioners incorporated teledentistry to ensure continuity of care for patients. It was deemed satisfactory for over 75% of patients to feel more at ease when doing the consultation at home [26].

To combat the pandemic, Australia is the first country to implement stringent intervention measures to halt the epidemic escalation by imposing travel restrictions, social distancing, and tracking by the federal government [27]. The Australian Health Sector Emergency Response Plan was activated by the government to set up a coordinated system of consultation and oversight at both the federal and state levels [28]. The main focus of these responses was on restriction of travel and border control, which were consistently assessed and modified based on their changing circumstances. A 14-day self-isolation was mandated for all international arrivals [28]. These measures played a crucial role in controlling the spread, as most infections have been from overseas and in returning travellers [28]. The state of New South Wales (NSW) experienced the highest incidence of the first COVID-19 wave as of May 2020 [29]. The public health COVID-19-associated restrictions that were implemented reduced the scope of dental practice and prevented some people from accessing dental services [30]. In a UK study reviewing the experiences of dentists practicing during the pandemic, staff recalled emotional and leadership challenges, coping strategies, and positive experiences [31].

A few other studies have reported on the experiences of Australian dentists during the pandemic [32,33]. A survey by Hopcraft and McGrath [33] identified the high levels of stress and burnout experienced by dentists, which were exacerbated by the pandemic. An online survey by Nahidi and Li [32] included a qualitative component for dentists to identify the challenges and some of the positive outcomes from their experiences in the pandemic. However, these earlier studies [32,33] focused on the perspectives of dentists who largely practiced in a private clinic and did not adequately capture the experiences of dental staff in public dental services, which continued to operate during the pandemic [34]. This study aimed to understand the perceptions of public dental staff towards providing dental care during the COVID-19 pandemic. The research questions included:What are the experiences of public dental staff towards providing treatment during the COVID-19 pandemic?What are the barriers and facilitators for public dental staff providing dental treatment during the pandemic?What are the supportive strategies that could assist public dental staff in providing dental treatment during the COVID-19 period?

The information gained from this study could assist health service providers and policymakers in developing more targeted and nuanced policies and protocols to better support the dental health workforce (particularly the public sector) and health consumers in the event of future pandemics.

## 2. Materials and Methods

### 2.1. Study Design

A qualitative descriptive research design was used to explore the experiences of dental staff through focus groups. A purposive sampling strategy was utilised to recruit dental staff to participate in focus groups.

### 2.2. Setting and Recruitment

Dental staff who practiced in public dental services across two Local Health Districts in Sydney, Australia, during March–June 2020 were eligible to participate in this study.

Flyers and an information sheet were emailed (by the relevant health service) to all eligible dental staff. Interested participants were asked to contact the study investigator for further information. Participants were then allocated into various focus groups depending on their availability. All attempts were made to ensure homogeneity of the focus groups by allocating a similar number of participants and skill mix (such as dental officers, dental assistants, and dental/oral health therapists). Focus groups were conducted until data sufficiency was reached and no new data was being generated.


*Inclusion criteria*


Dental staff (including dental officers, dental/oral health therapists, and dental assistants);Provided dental care in a public health setting across the two health districts between March 2020 and June 2020.


*Exclusion criteria*


Dental staff who were not working across the two health districts between March 2020 and June 2020;Administrative staff as they did not provide direct care to patients.

Ethics approval for this study was obtained from the South Western Sydney Local Health District Human Research Ethics Committee (approval number: 2020/ETH01902).

### 2.3. Reflexivity

Two authors (AG and AK) facilitated the focus groups. At the time, AG was a professor and a trained dentist with dental public health research expertise (MPH, PhD). AK was a research officer with public dental health research experience and expertise (PhD). Both researchers had experience in qualitative research, including conducting focus groups and integrated dental research, and had previously worked on projects with both health districts.

### 2.4. Data Collection

Five focus groups/one interview were conducted with 24 dental staff between March 2021 and May 2021 during an agreed time with the health districts. Before data collection, study information sheets were circulated to all interested participants, and informed written consent was obtained. No non-participants were present in the focus groups. Considering restrictions associated with physical distancing, the focus groups/interviews were conducted through an online teleconferencing platform (Zoom/Skype). All focus groups/interviews were recorded and were between 30 and 45 mins long. No repeat focus groups or interviews were conducted. Demographic information, including age, years of dental experience, number of dental professionals at their clinic, and whether they received training around COVID-19-associated dental protocols, was collected. Several questions and topics, which have been informed by the literature [33,34,35], were used to guide the focus groups and interviews, including:How have you been affected by COVID-19?How has your dental treatment planning changed?What protocols have been put into place at your service?What are your thoughts about providing treatment during COVID-19? (barriers/facilitators, safety, risk of infection);Use of remote dentistry;Need for any additional training/equipment or implementation of policies.

After each focus group, the two facilitators debriefed to discuss initial impressions with participants and to identify common experiences that were shared from the focus group and whether this was also identified in other focus groups. This debriefing process assisted the facilitators in determining the richness of the data collected and whether a moderately high degree of data sufficiency was being reached [35].

### 2.5. Data Analysis

The audio recordings of the focus groups/interviews were professionally transcribed to generate transcripts. The transcripts were analysed thematically using a primarily deductive (top-down) framework as described by Braun and Clark [36]. All participants were assigned pseudonyms to ensure confidentiality. Analysis was completed by two authors (Ak and TPN), who read all the transcripts for familiarisation with the data before conducting independent analysis using NVivo software version 13. Using qualitative analysis software Nvivo also assisted with data management and maintaining an audit trail. As part of the second phase (generating initial codes), initial codes were generated, followed by clustering similar codes together to create initial categories. The topic guide provided the framework for the areas that the research team aimed to explore in more detail; however, codes that did not fit into any of the categories in the initial framework were included in a miscellaneous category. For the third phase (searching for themes), these categories were combined to form themes across the focus groups. The two sets of themes, developed by the two authors, were reviewed for phase four (reviewing themes) and discussed with the larger team until a consensus was reached in the classification and naming of themes and sub-themes for phase five (defining and naming themes). For the final phase (producing the report), a final report was developed for the team to review, which identified the themes, subthemes, and quotes to support the thematic framework as well as answer the research aim. The framework and analysis were not returned to participants for comment or review; however, participants could opt-in to receive a final copy of the report. Demographic data was entered into a statistical software package (SPSS version 29) and analysed using descriptive statistics.

### 2.6. Rigour

A variety of strategies were employed to enhance the trustworthiness of the data collection and analysis [37]. For credibility, one author (AG) was involved in the recruitment of participants and organisation of focus groups. This allowed for lines of rapport, trust, and communication to have been built prior to the focus groups. Dependability and accuracy were addressed by the use of a professional transcription company to assist with converting the audio recording into verbatim transcripts. Member checking of transcripts was not deemed feasible for participants due to the time constraints of their clinical workload. Transcripts were reviewed by two authors (removed for peer review). Each author conducted a thematic analysis independently, and then a coding meeting was held to discuss and confirm the coding structure and preliminary findings. For transferability, detailed information on the recruitment and interview process, analysis, and findings, including direct quotes from participants, have been provided. To maintain confidentiality, all participants have been assigned pseudonyms denoted as P (1–24). To protect the identity of the health districts (HD), each site will be referred to as HD1 or HD2.

## 3. Results

A total of 24 participants (4 males and 20 females) from Health District 1 (n= 14) and Health District 2 (n = 10) were recruited. The majority of the participants were Australian-born (n = 16) and the mean age of all participants was 40.64 years (SD = 12.76) ranging from 19 to 63 years. The participants included a range of oral health professionals, including dental officers (n = 6), senior dental officers (n = 3), dental assistants (n = 10), and dental/oral health therapists (n = 4). The participants worked across various dental settings with different profiles and characteristics, including community health centres and hospitals involving a range of staff members (range 5–98).

Four themes and eight subthemes were identified in this study (see Table 1).

### 3.1. Responding to Protocol Changes at the Service

#### 3.1.1. Managing Constant Changes in Policy “Management Did the Best That They Could…”

Frequent policy change was part of the dynamic pandemic clinical environment. As described by one participant, “*what happened was each day was different. We’d get in, and one minute we’re following this thing* [policy or guidelines], *then we’d come in the next day, and all of a sudden, ‘Oh, we have to do this*,’ and it’s like, ‘*We need it done now* [referring to implementing updated guidelines or policy].’” (P13, HD2).

Many participants also discussed how policy changes impacted their experience of work. As conveyed by one participant, “As time’s gone on, and as more information’s come out, and how things and processes and procedures have had to be changed, the risk matrix has changed…” (P2, HD2). Although participants were aware that “putting job steps in place for staff to be safe…” (P2, HD2) was important, some participants experienced “…confusion of what needed to be followed…” (P13, HD2), which led to uncertainty in practice because of the competing policies.

In saying this, most participants from both sites felt that management was doing the best they could to support and navigate staff through the evolving policy changes in the pandemic environment:

…*the management did the best that they could. When you’ve got some bomb just dropping on you, I think the decisions they made were the best that they could at the time, I think because it was new to them as well*…(P13, HD2)

Some participants commented that daily updates were helpful: “*they were really good at informing us and keeping us updated with the emails*….” (P2, HD2). For one staff member, who was not part of management that was involved in organising the daily updates, they reported that at times this was not an easy job:


*I get feedback of where things that need to be put into place from management, and then part of my role is to actually put some of that into place. It’s not always the easiest of jobs because you’re trying to protect everybody, even though sometimes it makes extra work for people.*
(P2, HD2)

#### 3.1.2. Managing Patient Responses to Policy Changes “Most of Them Now Are Getting Used to It…”

Many participants acknowledged the challenges of implementing policy changes for their patients. For some participants, this began at screening, where they found patients did not want to cooperate with screening protocols and were hostile or abusive to screening staff.

Additionally, as reported by one participant, staff would get “…upset patients”—who say like, “Yeah, I’ve got a test negative,” but don’t have anything to prove it on and we decline the treatment. We’re having to decline the treatment because we can’t see them without results…” (P14, HD1), which also contributed to challenges around patient management. Other participants also highlighted issues including mandating mask-wearing in the clinical setting, encouraging patients not to bring family members, and patients who were unconcerned by the pandemic’s significance as challenges to implementing policy.


*Well, we still get a lot of patients not wanting to cooperate with the screening questions… Most of them now are getting used to it… but there are still some that don’t see the need for it and we will still cop abuse from them.*
(P11, HD2)

Another population group that posed its own unique challenge was the paediatric population. Due to restrictions within dental settings at the time, children were not able to be seen in the clinic unless it was determined that they were experiencing a dental emergency. Some participants noted that this caused some frustration and anxiety for parents:


*Even if they [parent of the child] just wanted to come in for ease of mind, for someone to look at them because they didn’t want to just accept it through the phone kind of thing… we actually couldn’t offer them an appointment until we could determine that it was an emergency.*
(P7, HD2)

Further, a couple of participants also raised that parents were concerned for their child’s oral health and experienced difficulty scheduling appointments with multiple children at home due to the lockdown.

“*It made it a little bit difficult at times to try and arrange appointments where we can schedule it to work for the parents plus the patient plus us, so it was a little bit difficult in terms of that in case there was a family of three or something that needed to come in with that.*”(P7, HD2)

Conversely, many participants discussed experiencing patients who were accommodating and compliant with the ongoing policy changes*, “I think from the patient’s perspective, for them they don’t necessarily want to come in if they didn’t have to*…” (P13, HD2) and “ *we had quite a few patients who actually understood and were appreciative that we are actually calling them and letting them know that this is happening*…” (P16, HD1).

### 3.2. Adapting to the Impact of COVID in the Workplace

#### 3.2.1. Being Supported Within the Workplace: “We Actually Worked as a Team to Get Through”

Considering the challenging experience of working in a health service during a pandemic, many participants shared factors that helped with the impact of the restrictions in the workplace, including feeling supported by fellow staff, management, and the community, and this made them feel appreciated in their role.

“…*I got really positive feedback in the community. Like I had so many good compliments—people wanting to hug me.*”(P15, HD1)

For other participants, the acknowledgement of their work by peers and the community made them feel proud to be a healthcare worker: “*we were proud in what we were doing. A sense of pride was there that we are doing something for the community*.” (P14, HD1).

Receiving practical and mental support also significantly contributed to adapting to COVID in the workplace. For many participants, this stemmed from a sense of job security and, overall, less stress as patient and clinic loads were reduced significantly with the escalating restrictions.


*Because work was a lot less. We weren’t seeing patients, so it wasn’t very stressful at work. I think it was actually quite nice to come in.*
(P10, HD2)

As noted by one participant, while everyone else was in lockdown, they felt privileged to be able to leave their homes to come to work:


*I think for myself, when everything got shut down and it was only essential workers that were allowed out, I was actually grateful that I did work in a healthcare-related profession because you are allowed to go out to work when everyone else has to stay.*
(P10, HD2)

Additionally, some participants also commented that it was helpful to receive guidance from close by larger hospitals and that “*we actually worked as a team to get through*…” (P18, HD1). Policies such as stricter PPE regulations also made a few participants report that they felt “*a bit more safe*” (P10, HD2).

Some participants reported instances where they felt support could have been improved. This was centred around sick leave protocols and COVID.


*I actually had quite a few COVID tests in the last 12 months, and I used up my sick leave really quickly, ‘cause you’re taking at least 1–2 days off, waiting for the results.*
(P5, HD2)

Some participants also mentioned that the COVID restrictions affected the morale of staff. Many staff members commented that due to physical distancing requirements, staff were isolated from each other, which affected their ability to socialise.


*Because it’s such a big building, everyone had to social distance and keep away from each other. We couldn’t have more than two people in the room. If we were walking in the corridors, we had to have PPE and things on… Everyone had to kind of just keep away from each other.*
(P6, HD2)

Despite efforts from the workplace to boost morale with “*fun tasks to do, some competitions and things*” (P6 HD2), participants explained that the impact of these measures felt temporary:


*It was a novelty I suppose in a way in the first little while, but then after that, we ran out of [activities] to do and colouring-in competitions to participate in, and I think “flat” is a really good word for it.*
(P6, HD2)


*I think after a few weeks, we were just kind of stuck with nothing to do.*
(P6, HD2)

#### 3.2.2. Managing Challenges Within the Workplace

Many participants were aware of the greater risk of infection they would be exposed to at work, more specifically if they were deployed to COVID screening clinics outside the dental setting. For some staff, this resulted in feeling fearful of contracting the virus and isolating themselves from their family members:


*I guess that was really scary for me at first. I didn’t tell my children I was going there [COVID testing clinic]…I think that affected me more because I was trying to stay away from as many as I could ‘cause I didn’t know if I was going to contract it from someone coming into the clinic. I was there for six months.*
(P4, HD2)

Other staff members also reported being worried about contracting the COVID-19 virus from patients; however, there were a few staff members who felt confident with the screening process for COVID-19 and felt that appropriate screening and mask-wearing provided good protection:


*from our perspective, they made it pretty easy for us, because we just asked the question. If there was something that we were unsure of, we would just give them a mask, ask them to wait outside*
(P6, HD2)

Following the rollout of the vaccine, some staff members anticipated this may change practice, although a few noted that having the vaccine did not change practice:

“*Yeah, it’s okay. No, since we’ve received the vaccine, there hasn’t been any change. We’re not behaving or doing anything differently*”(P3, HD2)

Other challenges raised by staff included the management of infection control and PPE supplies, specifically in the early phase of the pandemic due to the increased demand for PPE, cleaning supplies, and supply issues:


*The issue we had here was the uncertainty about PPE; about could we get supplies; about when did we need to wear PPE; could we wear a mask a whole day and not per every patient as we used to? The different levels of PPE*
(P9, HD2)


*Cleaning. There was a lot of cleaning done where we had to wipe all the handles and all the chairs and all that. We had to do that, what, six seven times a day.*
(P15, HD1)

A few staff members discussed that despite increased infection prevention and control measures, the demand to provide dental treatment did not seem to abate, which made it difficult to manage patient loads.


*It’s just demand which always overwhelms supply…Even though there was a stage where patient numbers did decrease a bit because we were doing the remote dentistry, and some patients were taking notice not to come in unless they had to, but the patients were the same, the treatment needs were the same.*
(P9, HD2)

### 3.3. Modifying Dental Treatment Planning

#### 3.3.1. Changes in Remote Dentistry: “We Were Closed to Your Regular Patients… Only Open for Emergency and Acute Care”

Following the implementation of the lockdowns and closure of all non-essential services, dental services across both sites implemented a teledentistry model. As described by one participant:

…*We were closed to your regular patients. We were only open for emergency and acute care patients, which meant that the teledentistry was completed for these patients first to deem whether or not it was something that they could come in…or whether it was something that we could give advice over the phone*…(P9, HD2)

For the most part, across both health districts, participants described that the primary role of teledentistry was to screen their patients, complete an interview over the phone, and book an appointment depending on their level of urgency.

“*There was a screening of patients, and there was a tele interview that was done over the phone as well, and then we would see them initially as emergency patients*…”(P3, HD2)

The majority of participants discussed the benefits they experienced while using the teledentistry model. Many participants recalled that the strict prioritisation guidance used for teledentistry allowed for improved clinical prioritisation and treatment of acute and emergency cases.


*If there’s a patient that we thought, ‘Yep, you need to have treatment done straight away,’ we would’ve booked them in anyway, and for those that were definitely not urgent… then that would’ve been pushed to—it’s not urgent at all.*
(P13, HD2)

The prioritisation was also noted to help manage the flow of patients and was considered to be accepted by some members of the public:

…*we were contacting these patients and we did cull out a lot of non-genuine emergency patients, and the good patients who understood the situation, that was a big help*(P9, HD2)

It also allowed clinicians to follow up with clients who were borderline acute cases to see how they progressed a few days prior to engaging them in an appointment. This was also noted to be beneficial in the paediatric population. As reported by one participant, in pre-COVID times, the acute care clinic would include children with non-acute cases; however, screening via the teledentistry consult allowed for true acute cases and emergencies to filter through:


*By having the teledentistry stream and actually having the availability for parents to send photos through of the concern, it eliminated a whole lot of kids, having to leave the house… It also meant that we could free up those appointments for those children that actually needed to have treatment completed*
(P9, HD2)

There were mixed opinions on whether teledentistry would be useful outside a COVID or pandemic situation. One participant felt that teledentistry was not sustainable in the paediatric population, while another felt that having some form of teledentistry might benefit acute-care paediatric patients:

*I’m not sure how it would be implemented to the best of our ability, but particularly for those acute care patients, having parents be able to talk directly to a clinician and either use FaceTime, which we did use for some patients, or just send a photo of the concern would eliminate a lot of kids having to come in*…(P9, HD2)

#### 3.3.2. Managing Infection Control and PPE Supplies “We Had Strict Infection Control Procedures and PPE in Place”

The treatments offered to patients changed significantly following the introduction of restrictions and guidelines from key dental regulatory agencies and NSW Health.

Some participants reported feeling quite restricted with procedures they were able to do:


*Patient-wise, I guess what impacted was we were very limited in what we could do with treatment…I think at one point we couldn’t even use what we had is a triplex which blow air to dry the tooth, we couldn’t even use that at one point.*
(P6, HD2)

Other participants raised the issue of stricter infection control measures. This varied from measures used for patients including the use of hydrogen peroxide mouth rinses:


*We implemented hydrogen peroxide mouth rinse as directed by the ADA and we followed what they had for level one dental treatment completed during restrictions.*
(P3, HD2)

To take time to implement stricter wipe-down routines following the treatment of patients:


*When we are cleaning up, we have to do a two-stage wipe down. Wipe down as normal and then go in with a stronger wipe to wipe down the second time, every single surface*
(P12, HD2)

To enable staff to keep up with the changing demands of infection control measures, some participants reported that they had longer appointments, which allowed them to include time for carrying out extensive infection control tasks.


*When we started to reduce the restrictions a little bit more, we did have staggered appointments, so instead of having half an hour per patient, we did have an hour, which gave us the time to be able to do the treatment plus the extra wipe down for the dental assistants*
(P7, HD2)

However, as reported by one participant, over time, restrictions changed, and this led to changes in their practice, which made some patients question the safety of these measures:

… *as the time went on and I guess more information came out, things did change. I guess that might have had an effect on people’s understanding of what was happening and whether or not what we were doing was actually the right thing and the safest thing at that time.*(P9, HD2)

PPE shortages were also a significant area of discussion for many participants. Some participants described a variety of concerns, from a shortage in PPE causing rationing of supplies to others raising that staff were not being compliant with changing PPE restrictions:


*We did have to limit the supply of personal protective equipment we had because the stock from the main hospital dwindled a little bit, so that had to go under lock and key just to make sure that we still had enough stock available to provide patient care on a daily basis.*
(P9, HD2)

Another barrier raised by participants was the occlusive nature of the PPE garments. Some participants felt this impacted their ability to communicate with patients, and others described uncomfortable side effects:

*We had to wear this helmet which was really heavy, and it was causing a lot of the clinicians neck pain. I think after a lot of input from a lot of different staff, we finally changed to a plastic visor which was a lot more comfortable*…(P3, HD2)

One participant also noted that the PPE sometimes impeded their ability to work:


*I think some dentists may have said, ‘oh, because of the extensive amount of PPE, it was difficult to do procedures that last a long time or anything.*
(P13, HD2)

In terms of changes for staff members, a few participants identified that they had to implement staggered lunch breaks for staff to reduce the amount of staff in the break room at any given time.


*We went from being able to all sit in the lunch room and eat lunch at the same time to having to stagger our lunch breaks so that we could do it within the restrictions of the one person per four square meters.*
(P18, HD1)

### 3.4. Recommendations for Training and Implementation of Policies

#### 3.4.1. Mental Health and Morale: “Our Mental Health Is, Is a Big Thing… I Think Is Really Important …”

Looking after staff and boosting morale was a key recommendation from both sites. Many participants discussed some of the strategies, including “*We tried to get all the freebies we could for staff just to lift spirits and say thank you, you know and all of that*” (P17, HD1), “*I may have seen some videos with some dancing…lot of Tik Toks* (P15, HD1)”, and “*we had served (catered) lunches*” (P15, HD1), all in an attempt to “*kind of not being depressed about everything all the time*” (P16, HD1). Additionally, other participants raised that having avenues to discuss mental health was important:

…*checking and seeing how our mental health is, is a big thing…it’s making sure that we as a team are actually doing okay. Whether it is from management, whether it is from your co-workers, having that happen I think is really important* …(P16, HD1)

A couple of participants also felt that introducing rotation between clinical and administrative roles for staff who had been re-deployed would allow for mental health relief.


*Give us a bit of like on off, on off. Like that would give a good pace for all of the clinicians rather than like a few clinicians who are just clinical and a few clinicians who are just teleconsult.*
(P14, HD1)

#### 3.4.2. Broader Consultation “Consulting with the Broader Base Before They Come up with Things”

In line with improving morale and focusing more on staff members, some participants expressed that involving staff as part of a “*pandemic committee*” (P22, HD1) would have allowed for firsthand input from staff who were on the “front line” and therefore would have enabled more tailored recommendations or strategies.


*Because a lot of us that worked in the clinics have experienced the situation. Maybe get some of these people together, and say okay, “This is how we’re going to do it. If this happens again, this is how we’re gonna do it.*
(P10, HD2)

Further, other participants expressed that staff members should be consulted before any changes to policy: “*I think probably a consulting with the broader base before they come up with things might be good*” (P16, HD1)


*Yes, it’s all good to say, ‘We’re gonna do it this way,’ but when we started doing it that way, it was good to get feedback from the clinics that were doing it to say, ‘No, I don’t really think it’s gonna work that way.’ Still keeping to the scenario of all the different rules and regulations, but I think feedback from the people that were actually facing it all would’ve been a better way to solve it, I think.*
(P10, HD2)

## 4. Discussion

This is the first qualitative Australian study to provide insight into the experiences of clinicians in public dental services that adapted and continued to operate to provide urgent and emergency dental care, while most private dental clinics were required to temporarily shut down [32,33,38,39]. The experiences of public dental staff highlight the unique challenges facing the dental team, both management and clinical staff, in continually adapting to a dynamic pandemic clinical environment, especially at the early stage. Ongoing changes in policies around practice reflected the importance of providing protection to staff and patients against the spread of the virus and, at the same time, offering support to dental staff who were at the frontlines with patients. An innovation that emerged from the restrictions in social distancing and access to healthcare services was the broad implementation of a teledentistry model in the public health service, which has the potential to be implemented sustainably through the long term.

One of the key areas discussed by dental staff was the need to constantly adapt to new or modified policies for dental treatment planning and management. Updated public dental service policies were developed in tandem with the wider evidence-based guidelines and policies that were disseminated at a national [40,41] and state level [34] to restrict the spread of COVID-19. Similar to our findings, British healthcare workers acknowledged that upper management was faced with difficult decisions that needed to be made quickly [32]. In the UK, frontline healthcare workers explained that the constant changes in policies created confusion with healthcare services [42,43]. Similar findings were also reported by Canadian dental hygienists, who also experienced uncertainty around practice due to conflicting messages from practice regulators [44]. Other countries with no single overarching guidelines or regulations left dental staff having to filter information to implement practices they deemed feasible [45]. This seemed to be a double-edged sword, however, as detailed guidelines may be helpful to navigate through a dynamic pandemic clinical environment. In Australia, private dentists followed guidelines released by the national professional dental organisation (Australian Dental Association) but found it difficult to keep up-to-date with the latest recommendations [39]. Similarly, private dentists in Jordan reported that the lack of clinical practice guidance was challenging [46].

While the number of patients seen at public dental services did not increase during the pandemic, the heightened sense of workload and mental burden described by some participants may be attributed to the implementation of additional treatment protocols and infection control measures, which was similarly found in another sample of Australian dentists [39]. The findings are also consistent with the study done by Sozkes and Olszewska-Czyż, who found increased stress and anxiety experienced by dental professionals because of quick changes in protocols and patient care (22). Further, the results of the research on burnout in Brazilian dentists found high levels of emotional stress and the necessity of efficient coping mechanisms (23). In addition, in Germany, nurses who were immunocompromised were working in a private practice or had financial concerns and had higher associated distress scores during the pandemic [47]. These findings all highlight the importance for healthcare management staff and policymakers to consider strategies that not only mitigate the spread of disease but also ensure clinical staff have adequate psychosocial support within their role [48].

There were several strategies that participants in our study believed improved, or could improve, their sense of support and mental health. These strategies included working together to improve morale and cohesion; receiving positive feedback from the greater community; feeling safer and more protected using PPE; special recreational activities to boost waning morale; additional leave to support the need for COVID-related testing; creating avenues to provide feedback on policies that impacted frontline staff; and having a shared sense of resilience and purpose in their profession. Many of the strategies identified in this study highlighted the importance of peer support, open communication, endurance, and cohesion from within the team, and the need to provide continuous motivation [48]. Aughterson and McKinlay [42] observed that while fatigue and exhaustion were common among healthcare workers during the pandemic, the availability of support structures at work and home was important. Appropriate coping strategies may differ between individuals as people respond differently to stress, indicating the need to debrief with individuals separately throughout the pandemic [49]. Thus, while policies may not have the flexibility to fully cater to the needs of healthcare workers, it is important that clear lines of communication are open and that there are opportunities for staff to debrief with the team to facilitate a sense of trust and reduce the mental burden.

The pandemic also saw an accelerated shift towards the delivery of healthcare using telehealth platforms. Although conventional dentistry requires in-person diagnosis, treatment, and management, participants in this study acknowledged the benefits of teledentistry as a tool to triage and prioritise patients who may require more urgent care and increase accessibility for follow-up care. Participants reported that teledentistry was acceptable [50] and was particularly useful for paediatric populations. The uptake in teledentistry was seen internationally [51]. Similar to our study, Di Spirito and Amato [52] identified teledentistry could be used as an auxiliary service to conventional dental checkups, specifically for consultations, diagnosis, assistance, and monitoring of paediatric consultations. In Canada, many provincial and territorial organisations developed teledentistry guidelines underpinned by the government’s plan for virtual health care [53]. In a few instances, teledentistry was implemented as a triage model to facilitate remote consultations; however, little is known about implementing these models [53]. A review by Hung and Lipsky [54] emphasised the need for clear guidelines and protocols provided by governing dental bodies to implement teledentistry into services sustainably beyond the pandemic. In Australia, no existing teledentistry models or guidelines have been disseminated broadly to inform dental practice. At the same time, the COVID-19 pandemic saw an increased backlog in public dental service waiting times and could be one strategy to reduce waiting times [55]. It is important to point out though that many of these supportive strategies would be impacted by the differences in national and state approaches to the management of the pandemic across public and private practice. For example, the state of New South Wales in Australia opted for a strategic approach centred on testing and gradual easing of pandemic restrictions [56], while Victoria enforced strict lockdowns allowing urgent care services only [57].

### Strengths and Limitations

One of the strengths of this study was that multiple focus groups over two health districts were conducted. This offered a wider lens into how different public health services implemented policies into practice, which helped with the credibility of our findings. A main limitation of this study was that the perspectives of management staff were not explored, which would have provided additional insight into the challenges with policymaking or implementation. Another limitation was that feedback on the analysis could not be sought from participants due to scheduling and funding constraints.

## 5. Conclusions

This study explored insights into how dental staff in the public dental service responded and adapted to the dynamic pandemic clinical environment. While dental staff understood the need to respond quickly to the changes in policies and practice, this placed a significant mental burden, which was mediated through various strategies that improved their sense of support and motivation. In response to the pandemic, both services also implemented a teledentistry model to facilitate triage, consultations, and follow-up appointments. While dental staff from both services acknowledged the usefulness of this model during the pandemic, future research and policies would need to explore how to integrate teledentistry into the service over the long term and whether this model would offer a high level of efficiency and effectiveness comparable to conventional dentistry.

## Figures and Tables

**Table 1 ijerph-21-01451-t001:** Major themes and subthemes.

Themes	Subthemes
Responding to protocol changes at the service	Managing constant changes in policy: “*Management did the best that they could*…” Managing patient response to policy changes “*Most of them are getting used to it*…”
Adapting to the impact of COVID in the workplace	Being supported within the workplace: “*We actually worked as a team to get through*” Managing challenges within the workplace
Modifying dental treatment planning	Changes in remote dentistry: “*We were closed to your regular patients. We were only open for emergency and acute care*” Managing infection control and PPE supplies: “*We had strict infection control procedure and PPE in place*”
Recommendations for training and implementation of policies	Mental health and morals: “*our mental health is, is a big thing…I think is really important*…” Broader consultation: “*consulting with the broader base before they come up with things*”

## Data Availability

The data that support the findings of this study are available from the corresponding author, AG, upon reasonable request.

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
