# Peer review of "Provision of Public Dental Services During the COVID-19 Pandemic: Experiences of Dental Staff in Greater Western Sydney, Australia"

_ijerph, 2024, doi:10.3390/ijerph21111451_

Round 1
Reviewer 1 Report
Comments and Suggestions for Authors
Dear Authors,
The topic of the article is interesting, and the qualitative study was very well designed. I have only one question that I would ask you to explain:
Materials and Methods
You have five focus groups. How did you ensure and establish the homogeneity of the group? How did you recruit the participants into focus groups, random selection, or?
Reviewer 2 Report
Comments and Suggestions for Authors
Thank you for assigning me as an reviewer for this manuscript. Covid-19 had many effects on dental practitioners and also changed the protochols of service providers during the pandemic. Many dental practitioners suffered from the effetcs of Covid-19. Very well written and interesting manuscript which I recommend to be accepted after minor revisions.
Line 48 to line 54;
In this paraghraph the authors may include some information about the "COVID-19 and respiratory protection for healthcare providers" which was very essential knowledge during the Covid-19 days. The practicing dentists were recommended to wear protective equipments. There are also recommendations in COVID-19 infection and the broader impacts of the pandemic on healthcare workers.
Line 55 to line 68;
The effects of Covid-19 to working conditions of dentists explained by authors. Only Jordan and Australia examples were given. Its possible to search the literature and add scientific references about effects of COVID-19 Pandemic on Working Conditions of Dentists in Poland and Turkey.
Line 55: Anxiety levels of dental practitioners in china was referred (14) in line 55 to 57. Other studies may be referred such as; anxiety Levels among Polish and Turkish Dentists during the COVID-19 Pandemic, Factors associated with depression, anxiety and stress among dentists during the COVID-19 pandemic, Burnout among Brazilian Dentists during the COVID-19 Pandemic: A Cross-sectional Study. to enrich the introduction part with wider perspectives.
Reviewer 3 Report
Comments and Suggestions for Authors
Providing public dental services during the COVID-19 pandemic: Experiences of Dental staff in Greater Western Sydney, Australia
Reviewer report
Title:
· Using ‘Provision’ instead of ‘Providing’ provides a more academic title.
Abstract:
· Line 19: The subheading should be revised as Background and aim.
· Line 25-27: What is stated between these lines should be moved to Methods.
· ‘teledentistry’ should be added to the keywords.
Introduction:
· It is insufficient and should be expanded a little more. As a suggestion, literature information could be added that includes complaints and symptoms of patients who applied to the dentist during Covid-19 and various dentist approaches to them.
· Why was this study needed? What are its potential contributions? What gap in the literature was it intended to fill? It should be clearly stated.
· Information about teledentistry should be given.
· At the end of the introduction, clearly re-state the aim of the study. If possible, state a hypothesis if it can be formed.
Materials and methods:
· Line 100: A study without exclusion criteria is unthinkable. Please state the exclusion criteria clearly.
· How was the sample size determined?
· Was informed consent obtained from participants?
· Ethics committee approval should also be stated here.
· How were the analysis and descriptive statistics of the data performed? Specify under a separate subheading.
Results:
· Revise Table 1 according to journal template.
· Research questions for all results given should be stated in the Materials and Methods section.
· This section is very complicated. It is quite difficult to understand. Please revise it in a more systematic academic fluency.
Discussion:
· The results have not been adequately related to the literature. Please highlight all of your results that agree and contradict the literature, with possible justifications.
Conclusion:
· This section is sufficient and appropriate.
References:
· Revise according to the journal template.
Comments on the Quality of English LanguageMinor editing of English language required.
Reviewer 4 Report
Comments and Suggestions for Authors
This is a good report, but there are some points that need to be addressed.
Introduction
- No reference to Covid-19 control measures taken in Australia.
- No reference to oral manifestations of COVID-19.
Methods
2.4 report names or initials instead of xx
3.1 improve the syntax and the language
Discussion
The difference in national environment and behavior, rules and laws, was not totally evaluated.
Comments on the Quality of English Language
Minor editing is needed
Round 2
Reviewer 3 Report
Comments and Suggestions for Authors
Thank to the authors for the revisions. Only minors:
There cannot be a study without exclusion criteria. Please ensure this.
References should be adapted according to the journal template.
Comments on the Quality of English LanguageMinor editing of English language required.
